# Exosome Liberation by Human Neutrophils under L-Amino Acid Oxidase of *Calloselasma rhodostoma* Venom Action

**DOI:** 10.3390/toxins15110625

**Published:** 2023-10-25

**Authors:** Suzanne N. Serrath, Adriana S. Pontes, Mauro V. Paloschi, Milena D. S. Silva, Jéssica A. Lopes, Charles N. Boeno, Carolina P. Silva, Hallison M. Santana, Daniel G. Cardozo, Andrey V. E. Ugarte, João G. S. Magalhães, Larissa F. Cruz, Sulamita S. Setubal, Andreimar M. Soares, Bruna Cavecci-Mendonça, Lucilene D. Santos, Juliana P. Zuliani

**Affiliations:** 1Laboratório de Imunologia Celular Aplicada à Saúde, Fundação Oswaldo Cruz, FIOCRUZ-Rondônia, Porto Velho 76812-245, RO, Brazil; suzanneserrath2@gmail.com (S.N.S.); as.pontes24@gmail.com (A.S.P.); paloschimauro@gmail.com (M.V.P.); millenamia02@gmail.com (M.D.S.S.); jessicah.amaral@gmail.com (J.A.L.); charlesboeno_chls@hotmail.com (C.N.B.); cacauusilvaa@gmail.com (C.P.S.); hallison.motta@gmail.com (H.M.S.); danielgomespvh@hotmail.com (D.G.C.); ugarteandrey@hotmail.com (A.V.E.U.); mcjoaogabrielmagalhaes123@gmail.com (J.G.S.M.); larissafaustina7@gmail.com (L.F.C.); sulamitasetubal@gmail.com (S.S.S.); 2Laboratory of Biotechnology of Proteins and Bioactive Compounds Applied to Health (LABIOPROT), National Institute of Science and Technology in Epidemiology of the Occidental Amazonia0 (INCT-EPIAMO), Oswaldo Cruz Foundation, FIOCRUZ Rondônia, Porto Velho 76801-059, RO, Brazil; andreimarsoares@gmail.com; 3Biotechonology Institute (IBTEC), São Paulo State University, Botucatu 01049-010, SP, Brazil; brucavecci@gmail.com (B.C.-M.); lucilene.delazari@unesp.br (L.D.S.); 4Graduate Program in Tropical Diseases and Graduate Program in Medical Biotechnology, Botucatu Medical School (FMB), São Paulo State University, Botucatu 18618-687, SP, Brazil; 5Departamento de Medicina, Universidade Federal de Rondônia, Porto Velho 76801-059, RO, Brazil

**Keywords:** Cr-LAAO, extracellular vesicle, snake venom, inflammation

## Abstract

L-Amino acid oxidase (LAAO) is an enzyme found in snake venom that has multifaceted effects, including the generation of hydrogen peroxide (H_2_O_2_) during oxidative reactions, leading to various biological and pharmacological outcomes such as apoptosis, cytotoxicity, modulation of platelet aggregation, hemorrhage, and neutrophil activation. Human neutrophils respond to LAAO by enhancing chemotaxis, and phagocytosis, and releasing reactive oxygen species (ROS) and pro-inflammatory mediators. Exosomes cellular nanovesicles play vital roles in intercellular communication, including immune responses. This study investigates the impact of *Calloselasma rhodostoma* snake venom-derived LAAO (Cr-LAAO) on human neutrophil exosome release, including activation patterns, exosome formation, and content. Neutrophils isolated from healthy donors were stimulated with Cr-LAAO (100 μg/mL) for 3 h, followed by exosome isolation and analysis. Results show that Cr-LAAO induces the release of exosomes with distinct protein content compared to the negative control. Proteomic analysis reveals proteins related to the regulation of immune responses and blood coagulation. This study uncovers Cr-LAAO’s ability to activate human neutrophils, leading to exosome release and facilitating intercellular communication, offering insights into potential therapeutic approaches for inflammatory and immunological disorders.

## 1. Introduction

L-Amino acid oxidase (LAAO) is an enzyme widely distributed in several species, such as insects, fungi, plants, and snake venoms [1,2,3,4,5]. The mechanism of action of LAAO is not yet fully elucidated but involves oxidative effects on various pathogens, such as fungi and viruses, as well as on various cell lines. Some of the observed effects are attributed to the production of hydrogen peroxide (H_2_O_2_) during oxidative reactions [6,7].

The literature demonstrates the biological and pharmacological effects induced by LAAO, including apoptosis induction [8], cytotoxicity [9], induction or inhibition of platelet aggregation [10], hemorrhage [11], hemolysis [12], bactericidal effects [13,14], leishmanicidal effects [15], edema [16], and activation of human neutrophils [17,18,19,20,21]. Many of the biological effects of LAAOs are attributed to the secondary effects of H_2_O_2_ produced during the enzymatic reaction [10,16,22,23].

The ability of LAAO to activate cells of the immune system, particularly human neutrophils, was reported by PONTES et al. [17,18] and PALOSCHI et al. [19,20,21] using an LAAO isolated from *Calloselasma rhodostoma* snake venom, named Cr-LAAO. This LAAO stimulates neutrophils chemotaxis, phagocytosis, production, and release of reactive oxygen species (ROS) through Nicotinamide Adenine Dinucleotide Phosphate (NADPH) oxidase complex activation, as well as the release of pro-inflammatory cytokines, myeloperoxidase, and lipid mediators. It also activates NLRP3 inflammasome [17,18,19,20,21].

The literature has demonstrated that Cr-LAAO is responsible for the activation of human neutrophils in vitro. This activation leads to the stimulation of the production of superoxide anion and H_2_O_2_ through the activation of NADPH oxidase. Additionally, it induces chemotaxis and phagocytosis [17,18,19]. Cr-LAAO also induces the activation of the corpuscle-dependent cyclooxygenase pathway lipids and cytosolic phospholipases A_2_, NLRP3 inflammasome, and the release of myeloperoxidase (MPO), interleukin 8 (IL-8), IL-6, IL-1β, tumor necrosis factor alpha (TNF-α), Leukotriene B_4_ (LTB_4_), Prostaglandin E_2_ (PGE_2_), and Neutrophil Extracellular Traps (NETs) [17,18,19,20,21]. Furthermore, it induces an acute response in vivo, as evidenced by leukocyte migration to the peritoneal cavity and the release of IL-6, IL-1β, LTB_4_, and PGE_2_ [24].

While leukocytes are implicated in several inflammatory pathologies, little is known about the state of activation of these cells during Cr-LAAO action. Recent research indicates that extracellular vesicles may be used by cells to transfer inflammatory mediators and receptors [25]. Majumdar et al. [26] demonstrated that these vesicles carry mediators such as LTB_4_, acting in an autocrine manner, sensitizing neutrophils, and amplifying the inflammatory response. Additionally, it was shown that these vesicles acted in a paracrine manner, promoting cell communication [27].

Exosomes, nanovesicles secreted by cells into the extracellular space, have an endocytic origin, and their size ranges from 40 to 100 nm in diameter [28]. Their formation occurs through the production of endocytic vesicles from the plasma membrane, called primitive endosomes. Subsequently, they become late endosomes, or multivesicular bodies (MVB), containing numerous intraluminal vesicles, which can be degraded by lysosomes or fuse with the cell plasma membrane, releasing their contents into the extracellular medium [29].

Through the interaction of exosomal membrane proteins with the target cell via juxtacrine communication, vesicles can initiate cell signaling events, activating the target cell [30]. Exosomes also hold potential as a novel therapeutic technique since they offer a method for delivering materials such as medicines, proteins, and siRNAs to specific cells or tissues through laboratory manipulation [31,32].

The ability to use exosomes as biomarkers, released under both normal and pathological conditions, has been demonstrated by the literature in various bodily fluids, including blood, tears, saliva, urine, and cerebrospinal fluid [33,34]. Recently, the importance of studying the intravesicular components of exosomes, such as proteins, mRNA, and miRNA, has been highlighted, as they are implicated in the progression of viral pathologies and their prognosis [35,36,37,38]. Moreover, its components can influence immuno-regulatory processes, such as tumor escape mechanisms, and regenerative and degenerative processes [39].

These vesicles are composed mainly of lipids and membrane proteins, similar to those present in the cell from which they originate. Furthermore, they may contain proteins and mRNA transcripts derived from the intracytoplasmic environment [40]. Their ability to interact with different cell types makes them an important mechanism for intercellular communication in the body. Furthermore, their role in the vascular environment depends on the cellular origin and the type of stimulus used to promote vesiculation, giving them specialized functions in their cellular targets [41,42].

The literature shows that exosomes are involved in cell–cell communication and can carry molecules such as MHC-II (linked to antigenic peptides), amplifying and modulating the immune response [43,44,45,46,47]. Additionally, exosomes are found in biological fluids such as plasma, bronchoalveolar lavage, breast milk, and saliva [48]. However, due to their small size and heterogeneity, detecting and classifying them remains a challenge [25]. Based on these considerations, studying the activation, formation, and release pattern of human neutrophil exosomes induced by Cr-LAAO may provide insights into their role in the inflammatory response and better characterize them for future clinical applications.

## 2. Results

The electrophoretic profile of exosome samples from RPMI (negative group), PMA (positive control group), and Cr-LAAO (experimental group) has revealed the presence of proteins at approximately 60 kDa under all conditions (Figure 1A). The size of the exosome vesicles present in the extracellular medium particles ranges from 8 to 70 nm for the negative control group, 6 to 50 nm for the positive control group, and 8 to 70 nm for the Cr-LAAO group (Figure 1B–D). Immunoreactivity of antibodies to CD63 and CD81 was detected in the samples of exosomes from neutrophils stimulated with Cr-LAAO, as well as in the negative control. However, cells stimulated with PMA did not exhibit immunoreactivity (Figure 1E). CD63 is a marker for extracellular vesicles in general, while CD81 is a specific marker for exosomes [49,50].

The proteomic analysis identified a total of 722 protein hits in all experimental groups. According to the maximum parsimony analysis, this study identified 330 proteins among the RPMI (negative control group), PMA (positive control group), and Cr-LAAO (experimental group). Detailed parameters for the identification of these proteins are provided in Appendix A.

Protein expression analyses revealed 206 common proteins among the three experimental groups. Specifically, 41 proteins were common between the control and Cr-LAAO groups, 81 proteins were exclusive to the control group compared to the Cr-LAAO group, and 28 proteins were exclusive to the Cr-LAAO group compared to the control group. Additionally, 9 proteins were common between the Cr-LAAO and PMA groups, 38 proteins were unique to the Cr-LAAO group compared to the PMA group, and 18 proteins were unique to the PMA group compared to the LAAO group. Finally, 30 proteins were common between the control and PMA groups, 89 proteins were unique to the control group compared to the PMA group, and 9 proteins were exclusive to the PMA group compared to the control group (Appendix A).

The Volcano Plot indicated proteins that were differentially expressed in Cr-LAAO-stimulated neutrophils compared to the control group. These proteins exhibited a fold change of ≥2 and a *T*-test analysis *p*-value < 0.05 (Figure 2A). Spectral count data for proteins from both groups were subjected to PLS-DA analysis, and the plot scores demonstrated a clear separation of the two groups and their respective biological replicates. This separation indicated high reproducibility among the replicates and a significant differential effect in the presence of Cr-LAAO (Figure 2B). In Figure 2C, the results of the VIP score analysis are displayed, highlighting the most important proteins associated with Cr-LAAO’s impact on neutrophils. Proteins with a VIP score ≥ 1.7 were considered statistically significant. Table 1 presents the up-regulated and down-regulated proteins in Cr-LAAO-stimulated neutrophils obtained through Volcano Plot Analysis.

To gain a better understanding of the biological processes, molecular functions, and cellular components associated with the differentially expressed proteins in exosomes from Cr-LAAO-stimulated neutrophils, gene ontology analysis was conducted. Among the proteins corresponding to biological processes, 16 proteins (27% of them) were involved in cellular processes; 9 (15%) in metabolic processes; 7 (12%) in localization activities; 6 (10%) in response to stimuli; 5 (8%) in biological regulation; 4 (7%) in the regulation of biological processes; 4 (7%) in immune system processes; 3 (5%) in the organization of cellular components or biogenesis; 2 (3%) in the upregulation of biological processes; 2 (3%) in developmental processes; and 2 (3%) in multicellular organizational processes (Figure 3A).

Regarding the proteins involved in molecular functions (Figure 3B), it was found that 33 proteins (48%) were related to binding functions, 24 (35%) to catalytic activity, 7 (10%) to structural functions, and 5 (7%) to transport activation functions. In the cellular components category (Figure 3C), it was observed that 24 proteins (44%) were associated with cellular anatomical entities, 20 (36%) with intracellular regions, and 11 (20%) with components of protein complexes. Figure 4 is a graphical representation of information related to protein–protein interactions. These interactions involve proteins that were expressed differently in neutrophils after being treated with Cr-LAAO.

## 3. Discussion

In the present study, to identify the presence of exosomes in the extracellular medium, isolation was standardized by filtration through 0.45 and 0.22 µM nylon membranes, followed by ultracentrifugation. The precipitate was collected, and subsequently, a Western blot was performed using antibodies against surface protein markers (CD63 and CD81) present in these vesicles. Exosome isolation derived from stimulated neutrophils under different experimental conditions (control, PMA, and Cr-LAAO) was successfully obtained, consistent with the results published by Lasser et al. [48] and Majumdar et al. [26]. By stimulating neutrophils isolated from peripheral blood with fMLP, Majumdar et al. [26] observed bands corresponding to the intravesicular components of myeloperoxidase, metallopeptidase 9, and 5-lipoxygenase, as well as the surface markers found on exosomes.

Simultaneously, the size of the exosomes present in the extracellular medium was determined using dynamic light scattering (D.L.S) with a ZetaSizer, which measures particle size. In this study, particles ranging from 10 to 100 nm were obtained, demonstrating exosome heterogeneity. Yamashita et al. [51] identified a similar size and distribution pattern of exosomes in cell supernatant samples from murine melanoma after ultracentrifugation.

The Western blot assay confirmed that the vesicles were indeed exosomes, as they exhibited the characteristic tetraspanins, including CD63, on their cell surface. Tetraspanins contain four transmembrane domains and are responsible for forming networks with a wide variety of proteins, creating tetraspanin-enriched microdomains (TEMs) [52,53]. Tetraspanin molecules can associate with various adhesion molecules such as β1 integrin and receptors, performing functions like cell proliferation, adhesion, activation, and migration. CD63, the first tetraspanin characterized, was found to be present in all samples in this study.

Furthermore, CD61, known as β3 integrin, is also a cell surface protein belonging to the integrin family. Its relationship with exosomes is linked to the role played by β3 integrin in exosome biogenesis and release. Studies have shown that β3 integrin (CD61) is involved in forming integrin complexes on exosome membranes. These complexes may play a role in selecting and incorporating specific proteins into exosome vesicles during their formation. Additionally, the presence of CD61 on exosome membranes may influence their interaction with target cells through the recognition of specific ligands [54,55]. Data from the present study indicate that CD61 was expressed in exosomes from Cr-LAAO-stimulated neutrophils and the control groups, but there was no expression in the PMA-stimulated neutrophils group. The differences in cell activation induced by PMA and Cr-LAAO may occur through different mechanisms that trigger distinct cellular responses. These differences in activation mechanisms can be attributed to several factors, including variations in intracellular signaling pathways. These pathways have different targets and effects within the cell, resulting in diverse responses [56,57].

Regarding the protein–protein interactions among proteins differentially expressed in Cr-LAAO-stimulated neutrophils, several interesting associations were identified. Among these, proteins from the α and β tubulin family were observed. Deficiency in these proteins or dysfunction of microtubules can lead to various human conditions and disorders, including neurodegenerative diseases such as Alzheimer’s, Parkinson’s, and Huntington’s disease [58], as well as breast cancer [59] and pancreatic cancer [60].

Two immune system proteins, myeloperoxidase (MPO) and complement 3 (C3), were also detected. These proteins play vital roles in the inflammatory response and the body’s defense against infections. Studies have suggested that MPO may influence complement activity [61,62]. MPO can modify C3, leading to its activation and the release of C3 fragments, which in turn can trigger inflammatory responses and attract immune cells to the site of infection [63,64].

Myosin and actin are the primary contractile proteins involved in muscle contraction. Myosin also plays a crucial structural role in maintaining the organization of these proteins in the sarcomere. Disruption in the function of these proteins can result in diseases such as cardiomyopathy and muscle paralysis [65,66,67]. It is important to note that these proteins have specific functions and interactions in various cellular contexts and physiological processes. There is no direct interaction between all of them in a single system or cellular event. Each of these proteins has its specific functions and roles in different biological systems and processes.

Additionally, the proteomic data from exosomes derived from Cr-LAAO-stimulated neutrophils, in comparison to the control, revealed the presence of several other proteins, including:Desmin, encoded by the DES gene, represents a muscle-specific ɪɪɪ-type intermediate filament crucial for proper muscle structure and function [68];Serpin B4, encoded by the SERPINB4 gene, can act as an inhibitor of parasitic and bacterial proteases, influence cell apoptosis, stimulate cell proliferation, and suppress immune defense against tumors [1,69,70];Antithrombin ɪɪɪ, encoded by the SERPINC1 gene, is a major serine protease inhibitor in plasma that regulates the blood clotting cascade by neutralizing thrombin, thereby inhibiting thrombosis [71];Titin, encoded by the TTN gene, is present in cardiac and skeletal muscles and functions as a “bidirectional spring” that regulates sarcomeric length and performs passive tension adjustments as needed [72,73,74];Myomesin 1, encoded by the MYOM1 gene, acts as an elastic band to maintain the structural organization of the myofibrillar M band [75,76,77];Voltage-dependent anion-selective channel protein 1, encoded by the VDAC1 gene, is found in eukaryotic cells and mediates the traffic of small molecules and ions across the outer mitochondrial membrane. It also interacts with anti-apoptotic proteins [78,79,80];Carbonic anhydrase 3, encoded by the CA3 gene, plays an important role in the transport of carbon dioxide [81,82];Kininogen-1, encoded by the KNG1 gene, is involved in blood clotting, inhibits thrombocyte aggregation, and has pro-inflammatory and pro-oxidant properties [83];Immunoglobulin kappa constant is produced by B lymphocytes and is encoded by the IGKC gene [84,85];Aconitate hydratase, encoded by the ACO2 gene, catalyzes the conversion of 2-oxoglutarate into succinyl-CoA and CO_2_ in the tricarboxylic acid cycle. It is mainly active in mitochondria [86,87];Peptidase D, encoded by the PEPD gene, plays a role in proline recycling and may be a limiting factor for collagen production [88];Sarcalumenin, encoded by the SRL gene, has a high capacity but low affinity for calcium binding [89,90];Dihydrolipolysin succinyltransferase residue component of the 2-oxoglutarate dehydrogenase mitochondrial complex, encoded by the DLST gene, catalyzes the global conversion of 2-oxoglutarate into succinyl-CoA and CO_2_, primarily active in mitochondria [86,87].

Among the data, proteins related to exosome composition and involved in cell signaling were also detected, including the 14-3-3 proteins (zeta, epsilon, beta, eta, gamma, and theta), which are encoded by the genes YWHAZ, YWHAE, YWHAB, YWHAH, YWHAG, and YWHAQ, respectively. These regulatory proteins are primarily found in the cytosol and the extracellular environment. They have the capability to interact with over 200 signaling proteins, including kinases, phosphatases, transmembrane receptors, enzymes, structural and cytoskeletal proteins, proteins involved in transcriptional control, and proteins associated with apoptosis. Moreover, they have demonstrated effects on the inflammatory process and are involved in the regulation of cell proliferation, differentiation, and metabolism [91,92,93,94,95].

Peroxiredoxin-2, encoded by the PRDX2 gene, plays a role in cellular protection against oxidative stress by detoxifying peroxides. It also functions as a sensor of hydrogen peroxide-mediated signaling events and may participate in signaling cascades involving growth factors and tumor necrosis factor-alpha (TNF-α), regulating intracellular concentrations of H_2_O_2_ [96,97] which have a role in neutrophil activation induced by Cr-LAAO [17,18,19,20,21].

Histone H2B and H3.3 histone B, encoded by the genes H2BC15 and H3-3B in humans, are involved in the structure of eukaryotic cell chromatin. They play a role in the process of compacting and decompressing DNA, which is important for the formation of NETs [98,99] as observed in neutrophils stimulated by Cr-LAAO [17].

L-lactate dehydrogenase A chain and L-lactate dehydrogenase B chain, encoded by the LDHA, LDHB, and LDH genes, are primarily responsible for converting pyruvate into lactate and transforming NADH into NAD+. These proteins are related to the pyroptosis process [100,101,102,103,104].

Arginase 1, encoded by the ARG1 gene, is a key component of the urea cycle and induces a profound suppression of T cell proliferation and cytokine synthesis by neutrophils [105,106,107].

Table 2 displays the proteins identified in the proteomics data when comparing neutrophils stimulated by Cr-LAAO with those stimulated by PMA.

Proteins related to the composition of exosomes and involved in cell signaling were also detected, such as annexin A2, encoded by the ANXA2 gene in humans. This protein can bind to cell membranes in a calcium-dependent manner and is involved in a variety of cellular functions, including signaling processes, cytoskeletal regulation, and interactions with lipids [131,132].

Rab GDP dissociation beta inhibitor protein, encoded by the GDI2 gene in humans, plays a key role in the GTPase cycle of Rab proteins. It helps maintain Rab in its inactive GDP-bound state, contributing to the precise regulation of intracellular vesicular trafficking and membrane transport. This regulation is essential for normal cell function and for maintaining the structure and function of cell compartments [133,134].

Coronin-1A, encoded by the CORO1A gene in humans, plays a key role in regulating the actin cytoskeleton. It influences cell migration, phagocytosis, cell shape, and signaling pathways in different cell types, especially in cells of the immune system [135,136] which were observed in neutrophils stimulated by Cr-LAAO [17,18,19,20,21].

Talin-1, encoded by the TLN1 gene in humans, is a protein involved in the formation of focal adhesions, which are points of contact between the cell and the extracellular matrix. These adhesions allow the transmission of mechanical signals and regulate cell adhesion and migration [137,138,139].

In conclusion, the data from the present study revealed that Cr-LAAO activates human neutrophils in vitro, stimulating the production of extracellular vesicles called exosomes, which may play an important role in cell-to-cell communication and the regulation of inflammatory processes. Proteomic analysis revealed the presence of several proteins involved in fundamental cellular processes, such as the regulation of the immune response, muscle contraction, and blood clotting. These findings highlight the importance of investigating the role of exosomes and differentially expressed proteins in Cr-LAAO-activated neutrophils, both in the physiological and pathological context. Understanding these mechanisms can provide valuable insights for the development of new therapeutic strategies and biomarkers in inflammatory and immunological diseases.

## 4. Material and Methods

### 4.1. Chemicals and Reagents

Turk’s solution, RPMI-1640, L-glutamine, gentamicin, fetal bovine serum (FBS), and Histopaque 1077 were purchased from Sigma-Aldrich Chem. Co. (Saint Louis, MO, USA). Antibodies for CD63 and CD81 were purchased from Invitrogen (Carlsbad, CA, USA) Pierce™ Chromogenic Endotoxin Quant Kit was purchased from Thermo Fisher Scientific (Waltham, MA, USA). Salts and reagents used in this study were obtained from Merck Millipore (Darmstadt, Germany) with low endotoxin or endotoxin-free grades, lysis buffer (NH_4_Cl 150 mM, KHCO_3_ 10 mM, Na_2_-EDTA 0.1 mM, pH 7.2), and phosphate buffer saline (PBS: NaCl 137 mM, KCl 2.7 mM, Na_2_HPO_4_.2H_2_O 9.1 mM, KH_2_PO_4_ 1.8 mM, pH 7.4).

### 4.2. Cr-LAAO Isolation

The Cr-LAAO isolation was performed following the protocol published by Pontes et al. [17] and Paloschi et al. [20]. The presence of endotoxin in Cr-LAAO samples was determined using the Quant kit derived from Pierce™ Chromogenic Endotoxin. The Cr-LAAO preparation exhibited the presence of 0.1 EU/mL of endotoxin, which was within the acceptable threshold of 1 EU/mL [21].

### 4.3. Isolation and Activation of Neutrophils

Neutrophils were isolated from peripheral blood obtained from healthy self-donors aged 18 to 40 years. Donors provided informed consent before blood collection, and all procedures were conducted per applicable regulations. The study was approved by the Ethical Committee of the Center of Tropical Medicine Research (CEPEM, Rondônia, Brazil—approval number CAAE 77529817.8.0000.0011), and participants provided informed consent before participating in the study according to the method described by Pontes et al. [17] and Paloschi et al. [21].

### 4.4. Exosomes Isolation

For exosome isolation, 1 × 10^7^ human neutrophils were obtained from three donors (3 Ns) and were incubated for 3 h at 37 °C, in a humidified atmosphere with 5% CO_2_ with 100 µg/mL of Cr-LAAO (experimental group), 500 ng/mL of PMA (positive control) and RPMI without supplementation (negative control) in a final volume of 400 µL. After the end of the incubation, the samples were centrifuged at 400× *g* for 10 min at 4 °C to remove the cells. Protease and phosphatase inhibitors (1 μg/mL) were added to the supernatant and subjected to sequential filtration through 0.45 and 0.22 µm filters to retain cell debris and particles larger than 200 nm. Then, the samples were subjected to ultracentrifugation at 100,000× *g* for 3 h at 4 °C. The precipitate containing the exosomes was resuspended in 200 µL of PBS containing protease and phosphatase inhibitors (1 µg/mL). The homogeneity of exosome size was determined using Zetasizer Nano ZS90 (Malvern Instruments, Orsay, France). A total of 1 μL of each preparation (experimental group, negative and positive control groups) was diluted in 999 μL of PBS, and the parameters of size distribution and zeta potential (electronegativity) of the exosomes were analyzed at 37 °C according to Khashayar et al. [140].

### 4.5. Protein Profile of Neutrophil Supernatants

An aliquot of each homogeneity of the exosomes was taken to determine the protein concentration by the BCA colorimetric method (Bicinchoninic Acid Protein Assay Kit, Sigma-Aldrich, St. Louis, MO, USA). To evaluate the protein profile after ultracentrifugation, the samples (20 µg) were incubated with 0.02% bromophenol blue, 10 mM mercaptoethanol, and 10% sodium dodecyl sulfate (SDS) for 5 min at 95 °C for protein denaturation. Then, they were separated on a 12% (*w*/*v*) polyacrylamide gel (SDS-PAGE). In the electrophoretic run, a constant current of 90 volts was fixed and, between 10–15 °C, to avoid band distortions and, consequently, resolution problems. For comparison purposes, 5 μL of Molecular Weight Standard (10–230 kDa Biolabs) was added to each electrophoretic run. The gel was stained with Coomassie Blue, and the bands were visualized directly.

### 4.6. Western Blot Protein Expression

For protein expression analysis, the proteins were electrotransferred to the adsorbent 0.45 µm nitrocellulose membrane, previously hydrated in water for 2–3 min in a transfer buffer for 5 min. The membrane was blocked with 3% (*v*/*v*) bovine serum albumin (BSA) for 1 h. After three 5 min washing steps with TPBS (PBS and 0.1% Tween), the primary antibodies to CD63 and CD81 were diluted in the titration established by the manufacturer and dispensed on the membrane, followed by incubation for 18 h at 20 °C. The washing steps were repeated, and then a secondary antibody conjugated with peroxidase (Sigma-Aldrich) was added and incubated for 2 h. After the washing steps, the membrane was developed using diaminobenzidine (DAB), 1 M Tris HCl pH 6.8, and 30% H_2_O_2_ [19].

### 4.7. Proteomic Analysis

Protein bands of each sample from RPMI (negative group), PMA (positive control), and Cr-LAAO (experimental group) were excised and submitted to in-gel digestion protein according to Shevchenko et al. [141] using trypsin as a proteolytic enzyme. Then, peptides were extracted from the gel matrix, followed by desalting in homemade C18 stage tips. The peptides were analyzed by LC-MS/MS in triplicate in an EASY nLC 1000 coupled to an LTQ Orbitrap XL ETD mass spectrometer (Thermo Scientific) equipped with a nanoelectrospray ion source (Phoenix S&T) at the Mass Spectrometry Facility RPT02H at Instituto Carlos Chagas, FIOCRUZ (Curitiba, Parana, Brazil) according to Forrest et al. [142].

### 4.8. Mass Spectrometry Data Analysis

Raw mass spectrometry data in RAW format was submitted to PatternLab software [version 4.0.0.84] [143] to obtain protein identification. The main parameters used in this tool were the UNIPROT database (Taxonomy Homo sapiens; 74,788 proteins, 5 November 2019); trypsin enzyme; allowance for 2 stray cleavages; post-translational modification carbamidomethylation of cysteine residues; variable post-translational modification oxidation of methionine residues; and MS 40 ppm and MS/MS tolerance errors of 0.0200 ppm. The maximum FDR (False Discovery) rate was considered to be ≤1%. A matrix compatible with the MetaboAnalyst 4.0^®^ program (https://www.metaboanalyst.ca/ accessed on 4 February 2022.) was constructed using the proteomic data. The spectral counts were normalized for each protein identified by the weighted average of the duplicate of each sample [144]. Proteins that have been identified in less than 70% of the samples were excluded from the analysis. Partial Least Squares (PLS) was used as the main method of multivariate analysis. Only signals present in 80% of the samples were considered for the generation of statistical models. Proteins were subjected to enrichment analysis for Gene Ontology (GO) terms “molecular function”, “biological process”, and “cellular component” using Panther software (www.pantherdb.org (accessed on 4 March 2023)). Protein interactions were also investigated regarding their biological processes using the STRING software, version 12.0 (http://string-db.org (accessed on 15 March 2023)), using the basic parameters: cut-off score of 0.90, confidence as network edges, and PPI enrichment *p*-value of <1.0 × 10^−16^.

## Figures and Tables

**Figure 1 toxins-15-00625-f001:**
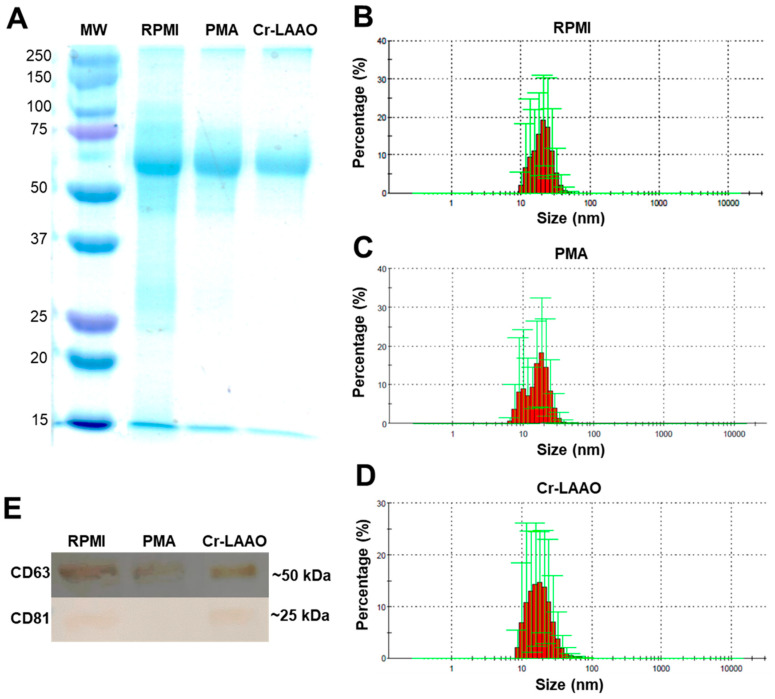
Exosome characterization. (**A**) Protein profile of the negative control (RPMI), positive control (PMA), and experimental group (Cr-LAAO) samples by SDS-PAGE 12% (*w*/*v*). MW: molecular weight. Histogram of the particle profile in the supernatant of the RPMI (**B**), PMA (**C**), and Cr-LAAO (**D**) groups after ultracentrifugation using dynamic light scattering (DLS) technique employed to determine the nuclear dimensions of the nanoparticles using the Zetasizer Nano ZS system (Mallvern^®^ Instrument, Malvern, UK). (**E**) Western blot for CD81 and CD63 using the supernatant of neutrophils (1 × 10^7^) stimulated with PMA (500 ng/mL) for the positive control (C+), RPMI for the negative control (C−), and Cr-LAAO (100 μg/mL) after 3 h of incubation in a humidified atmosphere of CO_2_ (5%) at 37 °C.

**Figure 2 toxins-15-00625-f002:**
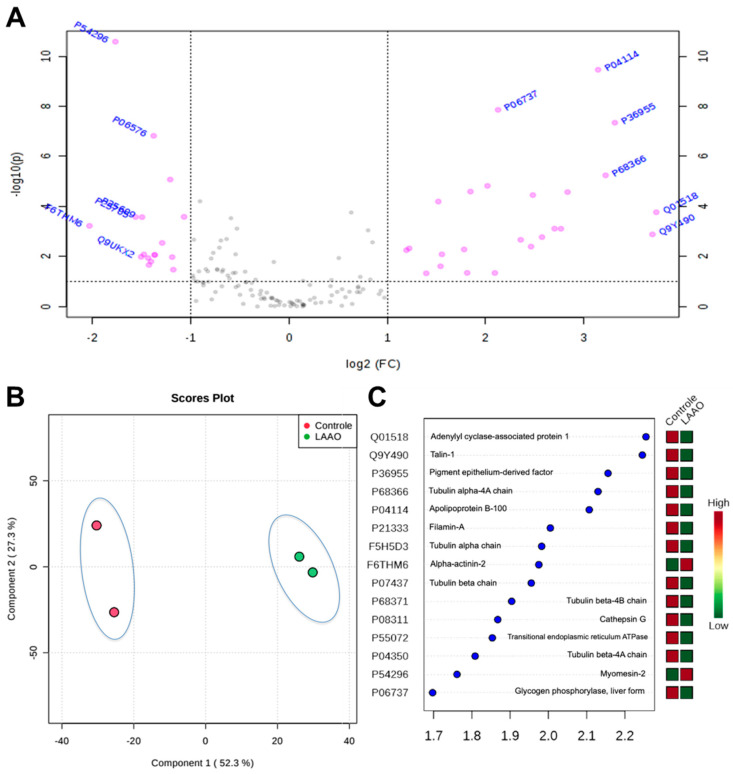
Volcano Plot analysis of Cr-LAAO – stimulated neutrophils. Important features selected by volcano plot with fold change threshold (x) and *t*-tests threshold (y). The pink circles represent features above the threshold. Note both fold changes and *p* values are log-transformed. The further its position away from the (0, 0), the more significant the feature is (**A**). PLS-DA analysis of differentially expressed proteins in response to Cr-LAAO treatment in cells (control group vs. LAAO group) (**B**). VIP score (Variable Importance in Projection score) represents the importance of each variable in the protein projection. The important factors identified by the PLS-DA analysis are listed in decreasing order of importance. VIP scores ≥ 1.7 were considered statistically significant. High VIP scores indicate a significant contribution of proteins to the separation between the control and Cr-LAAO groups. The boxes on the right indicate the relative concentration of the corresponding protein in each study group. Red boxes indicate higher protein abundance, while green boxes indicate lower abundance (**C**).

**Figure 3 toxins-15-00625-f003:**
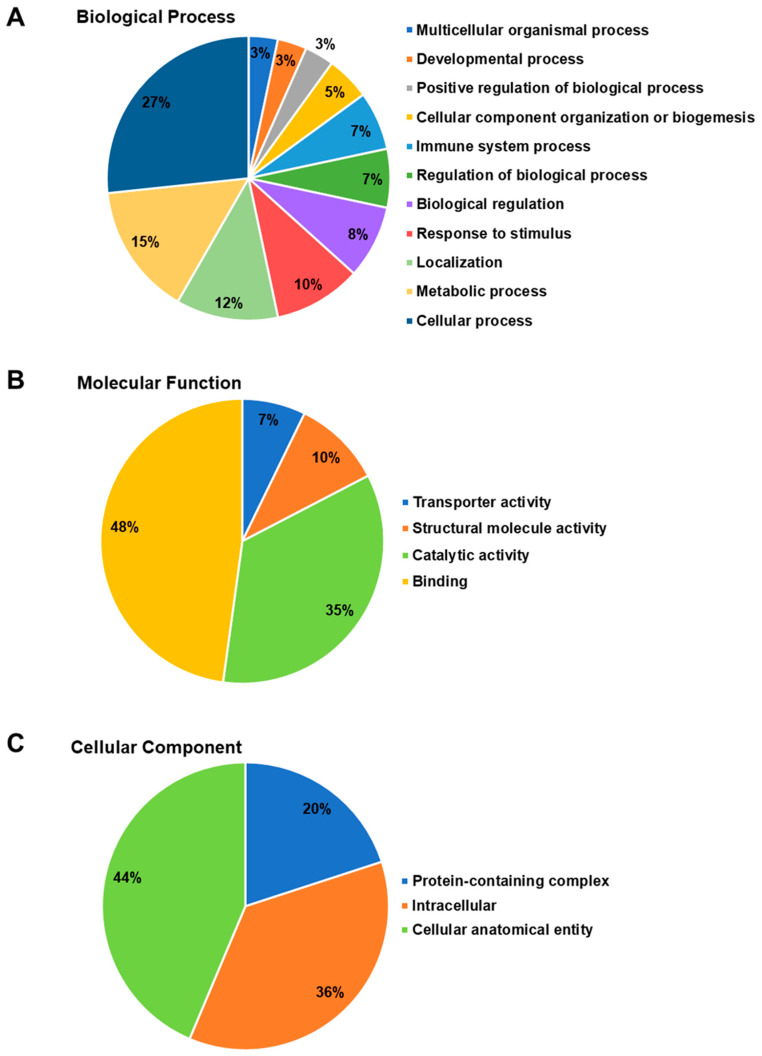
Ontology analysis of proteins differentially expressed in exosomes from Cr-LAAO-stimulated neutrophils. Human neutrophils were stimulated with 100 µg/mL Cr-LAAO for 3 h. After incubation, the supernatant containing the exosomes was collected and subjected to ultracentrifugation (100,000× *g*) for 3 h. The exosomes were subjected to polyacrylamide gel electrophoresis for protein separation and subsequent proteomics analysis. Data referring to the biological process (**A**), molecular function (**B**), and cellular component (**C**) were analyzed using Panther software. Unique proteins are present in the Cr-LAAO experimental group when compared to the control group.

**Figure 4 toxins-15-00625-f004:**
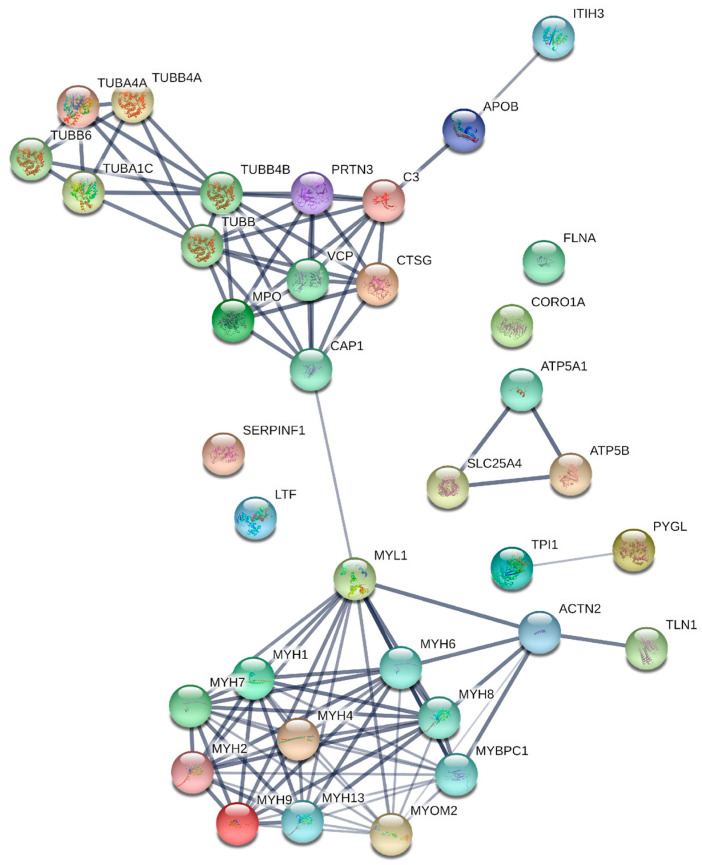
Protein–protein interactions of differentially expressed proteins in neutrophils upon Cr-LAAO treatment. Data were generated using String software (https://string-db.org, accessed on 4 February 2022). Circles denote proteins, and straight lines illustrate interactions between these proteins. The thickness of the lines corresponds to the strength of associations.

**Table 1 toxins-15-00625-t001:** Proteins differentially expressed in Cr-LAAO-stimulated neutrophils.

		Fold Change	
Protein Description	Access Code	LAAO/Control	Control/LAAO
Alpha-actinin-2	F6THM6	4.0756	-
Myomesin-2	P54296	3.3955	-
ATP synthase subunit alpha, mitochondrial	P25705	2.9427	-
Myosin-2	Q9UKX2	2.8325	-
Alpha-actinin-2	P35609	2.8138	-
Myosin-1	P12882	2.7773	-
Myosin-binding protein C, slow-type	F8VZY0	2.6963	-
Myosin-6	P13533	2.6811	-
Myosin-7	P12883	2.6417	-
ATP synthase subunit beta, mitochondrial	P06576	2.5931	-
Myosin-8	P13535	2.5761	-
Myosin-4	Q9Y623	2.5727	-
Alpha-actinin-3	A0A087WSZ2	2.4419	-
Myosin light chain 1/3, skeletal muscle isoform	P05976	2.3077	-
Myosin-13	Q9UKX3	2.2757	-
ADP/ATP translocase 1	P12235	2.2577	-
Sarcoplasmic/endoplasmic reticulum calcium ATPase 2	H7C5W9	2.0930	-
Putative elongation factor 1-alpha-like 3	Q5VTE0	-	2.2892
Lactotransferrin	P02788	-	2.3342
Coronin-1A	P31146	-	2.6324
Myeloperoxidase	P05164	-	2.866
Complement C3	P01024	-	2.9114
Immunoglobulin heavy constant gamma 1	A0A0A0MS08	-	2.9392
Tubulin beta-6 chain	Q9BUF5	-	3.4374
Myosin-9	P35579	-	3.5126
Myeloblastin	P24158	-	3.5948
Inter-alpha-trypsin inhibitor heavy chain H3	Q06033	-	4.0529
Triosephosphate isomerase	P60174	-	4.2686
Glycogen phosphorylase, liver form	P06737	-	4.367
Tubulin beta-4A chain	P04350	-	5.1224
Transitional endoplasmic reticulum ATPase	P55072	-	5.5079
Cathepsin G	P08311	-	5.5802
Tubulin beta-4B chain	P68371	-	5.9516
Tubulin beta chain	P07437	-	6.5033
Tubulin alpha chain	F5H5D3	-	6.8008
Filamin-A	P21333	-	7.1256
Apolipoprotein B-100	P04114	-	8.8368
Tubulin alpha-4A chain	P68366	-	9.317
Pigment epithelium-derived factor	P36955	-	9.9446
Talin-1	Q9Y490	-	12.962
Adenylyl cyclase-associated protein 1	Q01518	-	13.314

**Table 2 toxins-15-00625-t002:** Proteins differentially expressed in Cr-LAAO compared to PMA-stimulated in exosomes neutrophils.

Protein	Mechanism	Reference
Glucose-6-phosphate1-dehydrogenase; 6-phosphogluconate dehydrogenase; glucose-6-phosphate isomerase; transketolase.	Involved in the pentose phosphate pathway and plays an important role in the generation of NADPH.	[108,109,110,111,112]
desmoglein-1	Key role in the structure and function of epithelial cells, like cell–cell adhesion.	[113,114,115]
Myomesin 2	Formation and maintenance of the structure of the sarcomere.	[67,116]
Polyubiquitin C	Directly related to protein degradation through the proteasome.	[117,118,119]
Serpine B12	A protein involved in the downregulation of endopeptidase activity, providing protection to epithelial cells.	[120]
Filamin C	Organization and stabilization of the cytoskeleton, mainly in muscle cells.	[121,122]
Nebulin	Acts mainly in the regulation of muscle structure and function, it is crucial for the proper functioning of skeletal muscles.	[123,124]
Leukotriene A_4_ hydrolase	Plays a central role in the synthesis of leukotrienes, which are an important inflammatory mediator for immune response.	[125,126]
Vitamin D binding protein	Plays a crucial role in the transport and regulation of vitamin D in the body.	[127]
Cathelicidin antimicrobial peptide	Plays a key role in host defense against microbial infections, including bacteria, viruses, fungi, and even parasites.	[128,129]
Fibronectin	Plays multiple roles in cell adhesion, migration, wound healing, embryonic development, and immune response, produced by various cells, including fibroblasts, hepatocytes, endothelial cells, and muscle cells.	[128,130]

## Data Availability

The data presented in this study are available on request from the corresponding author. The data are not publicly available due to confidentiality reasons as they are stored in the Fiocruz Rondônia archives.

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
