# Peer review of "Exosome Liberation by Human Neutrophils under L-Amino Acid Oxidase of *Calloselasma rhodostoma* Venom Action"

_toxins, 2023, doi:10.3390/toxins15110625_

Round 1

Reviewer 1 Report

The study of the individual action of purified snake venom toxins contributes to understand their mechanism of action, impact on snakebite pathologies, therapies and also for their multiple applications as biotechnological and research tools.  In this scenario, the present study is relevant and add novel pieces on the toxinology puzzle of LAOOs. The manuscript brings interesting results with a well-written discussion. I have few comments.

1. Authors must include the key contributions of this study (please see journal guidelines). 

2.  Please revise the format of references according to the journal guidelines. 

3. Quality of figure 2 should be improved. 

4. Figures 3, 4 and 5 can be presented as figures 3 A, B and C. They are presented in discussed together. With this in mind, I believe this can facilitate understanding and improve the presentation of the results. 

5. Figure 6 was not mentioned in the results section. 

6. Please revise the title. 

7. The discussion is informative, but is too long. Authors must try to summarise focusing on their results. 

Author Response

Dear reviewer,

            First, I would like to thank you for your time and for the valuable comments made in relation to our manuscript. Below I will respond to each of your recommendations.

Answer to Reviewer 1

  1. Authors must include the key contributions of this study (please see journal guidelines).

Thank you for your consideration. The "key contributions" section has been included.

  1. Please revise the format of references according to the journal guidelines.

Thank you for the observation. The references have been formatted correctly.

  1. Quality of figure 2 should be improved.

            Thank you for the observation. We have adjusted the figure to one with higher quality.

  1. Figures 3, 4 and 5 can be presented as figures 3 A, B and C. They are presented in discussed together. With this in mind, I believe this can facilitate understanding and improve the presentation of the results.

Thank you for the suggestion. We have combined the three figures.

  1. Figure 6 was not mentioned in the results section.

Thank you for the observation. We have referenced the figure in the text.

  1. Please revise the title.

Thank you for the suggestion. We have changed the title.

  1. The discussion is informative, but is too long. Authors must try to summarise focusing on their results.

            Thank you for your consideration. We have made adjustments to the discussion.

Reviewer 2 Report

I have read manuscript “Type of the PaperExosome liberation by human neutrophils under Calloselasma rhodostoma venom action” and make the following recommendations:

Keywords: Use words different from those in the title so that a researcher interested in the topic has a greater chance of finding the manuscript. Ex: snake venom, amino acid oxidase…

Introduction:

Lines 53-55: the sentence requires reference(s): “This LAAO stimulates neutrophils chemotaxis, phagocytosis, production, and release of reactive oxygen species (ROS) via NADPH oxidase complex activation and proinflammatory cytokines, myeloperoxidase, and lipid mediators’ liberation as well as NLRP3 inflammasome activation”.

Discussion:

Line 187: Change “Calloselasma rhodostoma, (Cr-LAAO)” to CR-LAAO.

Line 189: Chnage “hydrogen peroxide” to H2O2

References:

Please read “Instructions for Authors”

“References must be numbered in order of appearance in the text (including table captions and figure legends) and listed individually at the end of the manuscript.” Please modify it

Author Response

Dear reviewer,

            First, I would like to thank you for your time and for the valuable comments made in relation to our manuscript. Below I will respond to each of your recommendations.

Answer to Reviewer 2

Keywords: Use words different from those in the title so that a researcher interested in the topic has a greater chance of finding the manuscript. Ex: snake venom, amino acid oxidase…

  Thank you for the suggestion. We have changed the keywords.

Introduction:

Lines 53-55: the sentence requires reference(s): “This LAAO stimulates neutrophils chemotaxis, phagocytosis, production, and release of reactive oxygen species (ROS) via NADPH oxidase complex activation and proinflammatory cytokines, myeloperoxidase, and lipid mediators’ liberation as well as NLRP3 inflammasome activation”.

Thank you for the observation. We have included the appropriate references.

Discussion:

Line 187: Change “Calloselasma rhodostoma, (Cr-LAAO)” to CR-LAAO.

Thank you for the observation. We have made the changes.

Line 189: Change “hydrogen peroxide” to H2O2

            Thank you for the observation. We have made the changes.

References:

Please read “Instructions for Authors”

“References must be numbered in order of appearance in the text (including table captions and figure legends) and listed individually at the end of the manuscript.” Please modify it

            Thank you for the suggestion. We have made the changes.

Reviewer 3 Report

Review of the article: Exosome liberation by human neutrophils under Calloselasma rhodostoma venom action

Manuscript ID: toxins-2612832

In this study, the authors investigated the activation patterns, formation, and proteomic content of exosomes released by neutrophils stimulated with RPMI, PMA, or Cr-LAAO. The differential expression of the proteins sheds light on potential mechanisms underlying the effects induced by Cr-LAAO on neutrophil behavior, which may offer insights into the cellular processes modulated by Cr-LAAO and highlight potential approaches for therapeutic development. I have some comments that the authors should notice, as bellows.

1.     The writing of some sections of the manuscript should be revised:

a.         The abstract needs to be more concise.

b.         I suggest that the first four paragraphs in Discussion be moved to Introduction, which needs to be further reorganized and integrated.

c.         The 9th to 14th paragraphs in Discussion describing the characteristics of diverse proteins are too wordy and should be presented in tables and discussed by focusing on the proteins such as those related to the therapeutic development of Cr-LAAO.

d.         According to Toxins regulations, the format of citations and references needs to be modified.

2.     Several citations in the text are not found or corresponding to those in the References. For example, ZULIANI et al., 2009 and IZIDORO et al. 2014 (at lines 35-36), ASWAD et al. 2016 (at line 219), Lasser et al. 2012 (at line 226), BOGGS et al. 2014 (at line 267), WANG et al. 2020 (at line 391), PUNWANI et al. 2016 (at line 406), PALOSCHI et al. 2022 (at line 436), Paloschi et al. 2022 (at line 444), Khashayar et al. 2017 (at line 460), and CHONG et al. 2014 (at line 504) are not found in the References. In addition, the references at lines 575, 654, 722, 726, 729, 751, and 755 is incomplete.

3.     What is the sample size of each control/group or trial?

4.     Spell out the full term of abbreviations/acronyms at its first mention.

5.     At line 2, “Type of the Paper” is redundant.

6.     At line 86, “from 8 to 70 nm” seems wrong according to Figure 2B.

7.     At Table 1, the decimal digits need to be unified.

8.     At line 233, “from 10 to 100 nm” is correct?

Author Response

Dear reviewer,

            First, I would like to thank you for your time and for the valuable comments made in relation to our manuscript. Below I will respond to each of your recommendations.

Answer to Reviewer 3

  1. The writing of some sections of the manuscript should be revised:
  2. The abstract needs to be more concise.

            Thank you for the suggestion. We have made the changes.

  1. I suggest that the first four paragraphs in Discussion be moved to Introduction, which needs to be further reorganized and integrated.

            Thank you for the suggestion. We have made the changes.

  1. The 9th to 14th paragraphs in Discussion describing the characteristics of diverse proteins are too wordy and should be presented in tables and discussed by focusing on the proteins such as those related to the therapeutic development of Cr-LAAO.

            Thank you for the suggestion. We have made the corrections.

  1. According to Toxins regulations, the format of citations and references needs to be modified.

            Thank you for the observation. We have made the changes.

  1. Several citations in the text are not found or corresponding to those in the References. For example, ZULIANI et al., 2009 and IZIDORO et al. 2014 (at lines 35-36), ASWAD et al. 2016 (at line 219), Lasser et al. 2012 (at line 226), BOGGS et al. 2014 (at line 267), WANG et al. 2020 (at line 391), PUNWANI et al. 2016 (at line 406), PALOSCHI et al. 2022 (at line 436), Paloschi et al. 2022 (at line 444), Khashayar et al. 2017 (at line 460), and CHONG et al. 2014 (at line 504) are not found in the References. In addition, the references at lines 575, 654, 722, 726, 729, 751, and 755 is incomplete.

            Thank you for the observation. We have corrected the references.

  1. What is the sample size of each control/group or trial?

            Thank you for your consideration. Samples from 3 donors (3 subjects) were used.

  1. Spell out the full term of abbreviations/acronyms at its first mention.

            Thank you for the observation. We have included the full term.

  1. At line 2, “Type of the Paper” is redundant.

            Thank you for the observation. We have corrected the redundancy.

  1. At line 86, “from 8 to 70 nm” seems wrong according to Figure 2B.

            Thank you for the observation. The presented values are right.

  1. At Table 1, the decimal digits need to be unified.

             Thank you for your consideration. We have unified the values.

  1. At line 233, “from 10 to 100 nm” is correct?

            Thank you for the observation. The presented values are right.

Round 2

Reviewer 3 Report

.